# Evaluation of the Field Efficacy of *Heterorhabditis Bacteriophora* Poinar (Rhabditida: Heterorhabditidae) and Synthetic Insecticides for the Control of Western Corn Rootworm Larvae

**DOI:** 10.3390/insects11030202

**Published:** 2020-03-24

**Authors:** Špela Modic, Primož Žigon, Aleš Kolmanič, Stanislav Trdan, Jaka Razinger

**Affiliations:** 1Plant Protection Department, Agricultural Institute of Slovenia, 1000 Ljubljana, Slovenia; primoz.zigon@kis.si (P.Ž.); jaka.razinger@kis.si (J.R.); 2Crop Science Department, Agricultural Institute of Slovenia, 1000 Ljubljana, Slovenia; ales.kolmanic@kis.si; 3Department of Agronomy, Biotechnical Faculty, University of Ljubljana; 1000 Ljubljana, Slovenia; stane.trdan@bf.uni-lj.si

**Keywords:** *Diabrotca v. virgifera*, inundative biological control, entomopathogenic nematodes, *Zea mays*, tefluthrin, thiacloprid

## Abstract

The western corn rootworm (WCR), *Diabrotica virgifera virgifera* LeConte (Coleoptera, Chrysomelidae), is an important insect pest of maize in North America and Central and Eastern Europe. In Central Europe, the larvae emerge in May and its three instars feed intensively on maize roots in June, causing plant lodging that leads to a loss of economic yield. A three-year field experiment (2016–2018) was conducted to compare the effectiveness i) of soil-applied granular insecticide based on the active ingredient tefluthrin, ii) of maize seeds dressed with thiacloprid, and iii) entomopathogenic nematodes *Heterorhabditis bacteriophora* Poinar (Rhabditida: Heterorhabditidae, product Dianem) against WCR larvae. An additional treatment with alcohol ethoxylate (i.e., soil conditioner) mixed with entomopathogenic nematodes was performed in 2017 and 2018 to check for any increase of entomopathogenic nematodes’ effectiveness. Field tests were carried out in two fields infested naturally with a WCR pest population, one in Bučečovci (Eastern Slovenia) and the other in Šmartno pri Cerkljah (northern Slovenia), exhibiting dissimilar pedo-climatic conditions and soil pest densities. The treatments were performed in five replicates per experiment in each year. The efficacy of the treatments was very similar at both locations, despite the approximately five-fold lower WCR soil pest densities in northern than in eastern Slovenia, as well as being constant over time. The largest number of WCR beetles was observed in the negative control, followed by that of beetles subjected to thiacloprid treatment (insignificant decrease taking into account the entire three-year dataset). Treatments with tefluthrin (44.1 ± 11.7%), *H. bacteriophora* (46.2 ± 7.4%), and *H. bacteriophora* + alcohol ethoxylate (49.2 ± 1.8%) significantly decreased the numbers of emerging beetles. Treatments of thiacloprid, *H. bacteriophora*, and *H. bacteriophora +* alcohol ethoxylate additionally led to significantly increased maize plant weights. Furthermore, entomopathogenic nematodes were able to persist in maize fields for almost five months at both experimental locations in silty and sandy loam soils. It was concluded that the control of WCR larvae in maize using the entomopathogenic nematode *H. bacteriophora* is as effective as a tefluthrin treatment, and could thus offer a sustainable *Diabrotica v. virgifera* biological control management option in Europe.

## 1. Introduction

The western corn rootworm (WCR), *Diabrotica virgifera virgifera* LeConte (Coleoptera, Chrysomelidae), was assigned the status of an insect pest of corn (*Zea mays* L.) in the early 20th century [1] and is still an important and challenging pest worldwide. Organochlorine insecticides were first used against WCR by farmers in the USA in 1949, but the pest developed resistance within five years [2]. After that, first carbamate and organophosphate, then pyrethroid insecticides were used for rootworm control, but evolution of the resistance of adult WCR to these insecticides has been reported [3,4,5]. Later, genetically engineered maize that produces crystalline toxins derived from the bacterium *Bacillus thuringiensis* (Bt maize) [6,7] was used in 2003 in the USA. However, in 2009, the first WCR population showing resistance to Cry3Bb1 corn and mCry3A corn was discovered [8]. Additionally, a WCR population with resistance to Cry34/35Ab1 corn was confirmed in 2016, which is a consequence of different mechanisms, including field-evolved resistance and cross-resistance [9].

In Europe, crop rotation, as an agricultural practice, remains effective as a tool for managing WCR [10,11,12]. However, in the western United States Corn Belt, crop rotation is considered to have limited value for WCR management because the species has lost its fidelity for corn and lays eggs in fields planted with other crops [13,14]. In addition to crop rotation, and soil or foliar insecticide applications are frequently used in the EU as management options against WCR, especially in fields with high population pressure. Use of foliar insecticides against WCR adults in Slovenia, as well as in some other European countries, is problematic due to (i) a lack of machinery for foliar application of pesticides in maize, (ii) dispersed arable land (small fields), (iii) prohibition of aerial application of pesticides, and (iv) difficulties in the official registration of proper efficacy pesticides. Seed coating and application of soil insecticides appears to be less invasive and is thus favored in maize protection programs [15]. Pyrethroid insecticides are one of the few chemical options for controlling corn rootworm larvae. Especially common is the use of the soil insecticide tefluthrin [16]. Tefluthrin is a synthetic pyrethroid insecticide targeting soil-dwelling pests and is applied in a granular form at the time of planting as an in-furrow treatment. The addition of tefluthrin mainly affects WCR larvae, resulting in a significantly lower adult emergence rate [17], reduction of maize root damage, and therefore reduced lodged plants [18]. Neonicotinoid insecticide seed treatments are commonly used in pest management systems because seed-applied insecticides can control various underground pests simultaneously. After the introduction in 2013 of restrictions on the use of some neonicotinoid active substances (Regulation (EU) No 485/2013), thiacloprid has been recommended for maize seed treatment, because of its lower toxicity to honey bees [19].

The development of behavioral and physiological insecticide resistance and withdrawal of many insecticides due to the non-target effects against soil-dwelling arthropods, together with the adverse effects on the environment and human health, have fueled research for environmentally friendly alternatives for WCR control. In Europe, various methods have been investigated for reducing the WCR population. For example, pheromone mating disruption using 8-methyl-2decanolpropanoat [20], hybrids possessing native resistance and tolerance to WCR [21], and the attract and kill strategy [22] have been developed, but their implementation is challenging and these methods have not been adopted by many growers.

Kuhlmann and van der Burgt (1998) [23] recommended that biological control options should be considered for WCR management in Europe. Laboratory screening tests of eight entomopathogenic nematode (EPN) species available in Europe showed their ability to infest the larval stages of WCR [24], among which the species *H. bacteriophora* was the most promising candidate as a biological control agent [25]. It reduced the population of the WCR larvae by up to 65% and the lodging of plants by up to 60% in field trials, which is comparable to the activity of soil insecticides [26,27,28]. *H. bacteriophora* was firstly determined in Slovenia in August 2008 [29]. After its discovery, Slovenia became one of the countries where the use of nematodes as a biological protection agent is sanctioned by law also for outdoors application. Establishment of the presence of entomopathogenic nematodes in Slovenia was first studied by Laznik and Trdan (2012) [30]. Besides inundative EPN use, the efficacy of entomopathogenic fungi (EPF) for WCR larvae management has been studied [31]. Field efficacy assessment of EPF, EPN, and insecticides revealed that EPF application shows lower efficacy than EPN or insecticides [28]. An evaluation of the potential non-target effects of these biological and chemical control agents was investigated by Babendreier et al. (2015) [32]. However, a more recent field study showed that a combination of entomopathogens with chemical insecticides performed better than either product alone [33].

Despite *H. bacteriophora*’s great potential as a biological control agent for controlling WCR larvae in the soil [12,25,34,35], its efficacy is often highly variable due its persistence, virulence, and survival in different fields that are affected by dissimilar soil and climatic conditions [36,37]. Besides their susceptibility to UV radiation [38], high temperatures and desiccation [39], and moisture condition is one of the crucial factors that affects nematode activity and survival in the soil [36]. Various commercial products have been developed to reduce surface tension and improve soil hydration, making more water available for plants. However, better distribution of water in the soil profile may also enhance the dispersal of EPNs and help prolong their survival [40]. The objectives of this study were thus threefold: (i) Comparison of the field efficacy of *H. bacteriophora* and chemical control options commonly used in maize for controlling WCR; (ii) investigation of the potential efficacy of an *H. bacteriophora* increase when blended with a soil conditioner (alcohol ethoxylate); and (iii) assessment of the persistence of *H. bacteriophora* in the soil.

## 2. Materials and Methods

### 2.1. Field Sites and Experimental Set-Up

The three consecutive years (2016–2018) of field studies were carried out in a maize-growing area in Eastern Slovenia at Bučečovci (site 1) and Northern Slovenia, Šmartno pri Cerkljah (site 2) located approximately 170 km apart (Table 1). Both field sites had a natural pest infestation of western corn rootworm (WCR) (site 1 since 2004, site 2 since 2008) and were for four years prior to our experimentation in monoculture production of maize.

Both fields were ploughed and harrowed before sowing. At site 1 corn seeds were sown by Monosem NC classic pneumatic planter, at an inter-plant spacing of 70 cm and an intra-row spacing of 16–17 cm, resulting in 85,000 plants per ha (Table 2). The maize seeds were sown at site 2 by a Gaspardo pneumatic planter using the same procedure. Each treatment comprised 4 rows, each 100 m in length. The middle two were used for assessments that were performed in 5 replicates along a row, separated by approximately 20 m. Each treatment had a surface of 0.028 ha; the total size of each experimental field was 0.14 ha. Weeds were controlled at both locations after the emergence of corn (BBCH 12–14), using herbicide Lumax (a.i. S–Metolachlor, Terbutilazin and Mezotrion) at a rate of 3.5 L/ha. In both regions, the flight dynamics of WCR adults were recorded using pheromone PAL traps (Csalomon, Hungary) within an official pest monitoring program. Traps were placed 1 km away from the experimental maize fields in mid-June and inspected weekly until October. Pheromone bait and sticky traps were replaced monthly.

### 2.2. Treatments

*Treatment with entomopathogenic nematodes (EPN).* The product Dianem^®^ is based on an entomopathogenic nematode hybrid of European and US strains of *Heterorhabditis bacteriophora* Poinar, which is produced by e-nema GmbH (Schwentinental, Germany). The free-living stage of *H. bacteriophora*, known as the infective juvenile, was shipped by the producer to the Agricultural Institute of Slovenia (AIS). Upon arrival, they were stored in the fridge at 7–9 °C in darkness until use. A week before application, subsamples of nematodes were taken to evaluate the quality of the shipment batches through assessment of the mortality of larvae of *Galleria mellonella* (Lepidoptera, Pyralidae). Three plastic cups (diameter 40 mm, height 60 mm) per nematode shipment batch were filled with 200 g of moist sterilized sand to which 5 larvae and 100 infective juveniles were added. A mortality of 80%–100% *G. mellonella* larvae was observed after one week in darkness and at 22 °C for all nematode batches. This was considered sufficiently high for testing. The infective juveniles were applied as liquid formulations, with application rates of 2 × 10^9^ infective juveniles per ha in 200 L of water. Suspension of the infective juveniles was applied at sowing, directly into the seed furrows, at a depth of 5–6 cm using a special EPN application system developed at the Agricultural Institute of Slovenia. The EPN application system consists of a 30-L tank, a 12 V pump, and distribution pipes. The tube for the application of EPN is aligned to each row unit on the planter, which allows the direct application into the vicinity of the sown maize kernel of the EPN suspension during sowing. The tractor speed at the time of nematodes application was 3 km/h. The seeds of maize hybrids LG 34.90 (Limagrain, France) and Chapalu (RWA AG, Austria) used in the experiments were treated with bird repellent Korit 420 FS (a.i. 42% ziram) and fungicides Flowsan FS (a.i. thiram) and Maxim XL 035 FS (a.i. fludioxonil and metalaxyl-M) at the dose 12.5 mL/50,000 seeds.

*Treatment of the EPN in combinations with alcohol ethoxylate.* Transformer^®^ (Oro Agri International Ltd., The Netherlands) is an alcohol ethoxylate-based soil conditioner, formulated to improve the distribution, penetration, infiltration, and retention of water in the soil, resulting in reduced water logging and run-off and increased root formation. It was applied at a rate of 150 mL per 30 L of water together with EPN (recommended dosage 5 L/ha in 1000 L water/ha, i.e., 0.5 % concentration).

*Treatment with thiacloprid.* Maize seeds were treated with the neonicotinoid thiacloprid (Sonido FS 400; Bayer AG, Germany) at a dose of 0.125 L/50,000 seeds.

*Treatment with tefluthrin.* The tefluthrin-based granular soil insecticide Force 1,5 G (Syngenta Crop Protection AG, Switzerland) was applied at a rate 12 kg per hectare into the seed furrows during sowing using a micro granular applicator.

*Control treatment.* Seeds without added insecticides were used.

### 2.3. Evaluation of Treatment Efficacy

The field efficacy of treatments was assessed by counting the number of emerged WCR beetles in adult emergence cages. In each treatment, five emergence cages, approximately 20 m apart, were installed over the inner two rows before adult emergence was expected in mid-June. Twelve whole uncut maize plants were covered by a one square meter gauze cage (1000 mm × 1000 mm × 2300 mm in size) in 2016 and 2017. The emerged beetles were caught using an unbaited Pherocon AM yellow sticky trap (Trécé, Adair, OK, USA), placed 1.5 m above the soil in the middle of the emergence cage. Only captured beetles on the yellow sticky trap were counted in 4-week intervals from the end of June to mid-September, at the same time as traps were replaced. In 2018, pyramidal emergence traps designed by Rauch et al. (2016) [41] were used as adult emergence cages. The installation of this type of trap is based on cutting maize plants at a height of 20 cm, shortly prior to the beetles’ expected emergence. However, in this study, instead of covering one maize plant with a single trap as according to Rauch et al. (2016) [41], 12 plants were covered within 1 square meter.

### 2.4. Evaluation of Maize Root Damage, Plant Lodging, Fresh Plant Weight, and Fresh Grain Yield Per Plant

The root systems (250 × 250 × 200 mm) of 10 maize plants were dug out from each treatment to measure root damage at the end of July in 2016, 2017, and 2018. Soil was initially removed from the root systems by gently shaking the roots and, secondly, using a high-pressure water spray. Damage was rated using the 0.00–3.00 node injury scale [42]. The suggested economic threshold level in conventional grain maize is 0.25 according to this scale. The percentage of lodged plants, or plants exhibiting typical WCR larvae damage (i.e., ‘goose necks’), was assessed in August by counting the symptomatic plants in the two middle rows of each treatment along the whole length of the experimental plots. Prior to harvest, in mid-September, the fresh plant weight and corn ears’ weight were measured. Five randomly selected whole plants with corn ears per replicate, thus 25 per treatment (*n* = 5), were cut at the stem base and weighed using a portable scale (DZD, model DJ30KL, G&G GmbH, Kaarst, Germany). The total number of corn ears per plant was counted and weighed separately to determine the fresh grain yield per plant.

### 2.5. Assessment of Nematode Persistence in Soil

In order to study the persistence of applied nematodes *H. bacteriophora* in the soil, soil samples were taken monthly from the time of application until September, at two depths 5–10 cm and 10–15 cm. Three 0.2-L soil samples were taken randomly with a garden spatula in the EPNs, EPNs with soil conditioner, and control treatment. Naturally occurring indigenous EPN populations at the experimental field sites were assessed in a control treatment. Each sample was put into a 250-mL covered plastic box and transported to the laboratory in a portable cool box. The presence or absence of nematodes was determined by using an insect baiting technique [43]. Three larvae of *Tenebrio molitor* (Coleoptera, Tenebrionidae) and three larvae of *Galleria mellonella* (Lepidoptera, Pyralidae) were placed in each cup containing a soil sample and used as bait for the EPNs. The mortality of the larvae was assessed during 2 weeks of incubation in a growth chamber in darkness at 21 °C and 77 ± 3% relative humidity. Dead insect larvae were placed into a nematode emergence trap [44] for another two weeks for potential propagation of nematodes.

### 2.6. Data Analysis

The number of WCR beetles caught on yellow sticky traps in emergence cages was analyzed separately for each year and location by ANOVA and Dunnett’s multiple comparison test. The entire three-year dataset was analyzed by the general linear model (GLM), where the effects of treatment (thiacloprid, tefluthrin, *H. bacteriophora*, *H. bacteriophora* + alcohol ethoxylate, control), location (site 1—Bučečovci, site 2—Šmartno pri Cerkljah), year (2016, 2017, 2018), and the interaction treatment × location were analyzed on the parameters: Number of emerged beetles, fresh plant weight, and fresh grain yield per plant. Further, Tukey’s honestly significant difference (HSD) procedure at the 95% confidence level was used to discriminate between the treatments within the three-year dataset. The analyses were performed with the statistical software Statgraphics Centurion XVI (StatPoint Technologies, Inc., The Plains, VA, USA) and GraphPad Prism 5.00 (GraphPad Software, Inc., La Jolla, CA, USA). The number of replicates (*n*) is indicated in the figure or table captions.

## 3. Results

### 3.1. Insect Pest Density at Two Experimental Sites

The average emergence of western corn rootworm (WCR) beetles per plant from pest-infested soil in the field cages in the control plots was approximately five-fold lower at site 2 (Šmartno pri Cerkljah, Northern Slovenia; 1.6 ± 0.3 in 2016 and 1.8 ± 0.2 in 2017) than at site 1 catches (Bučečovci, Eastern Slovenia; 7.5 ± 0.4 in 2016 and 9.2 ± 0.8 in 2017). The average catches in 2018 at site 1, where different emergence traps were used, were approximately two-fold lower than in 2016 and 2017 at site 1: 3.6 ± 0.0.

The official pest monitoring survey, where PAL pheromone traps (Csalomon, Hungary) were used to catch flying male beetles, showed similar catches near the two sites (Figure 1). The first specimens appeared each year of the experiments at the end of June while the peak of outbreak occurred at the beginning of August. WCRs were observed on individual fields until the end of October [45].

### 3.2. Treatment Efficacy

The general linear model and the HSD procedure showed a significant effect of treatment (F_4, 114_ = 4.73; *p* = 0.0015), year (F_2, 114_ = 40.62; *p* = 0.0000), and location (F_1, 114_ = 154.89; *p* = 0.0000) but not the interaction of treatment × location (F_4, 114_ = 1.79, *p* = 0.1368) on the number of emerged beetles. The highest number of WCR beetles was caught in the negative control, followed by the treatment with thiacloprid (75% of control beetle emergence; insignificantly different from control treatment beetle emergence considering the entire three-year dataset). Treatments with tefluthrin (56% control emergence), *H. bacteriophora* (54%), and *H. bacteriophora* + alcohol ethoxylate (51%) significantly decreased the average number of emerged beetles in three consecutive years and were statistically indistinguishable according to the GLM and HSD analyses (Table 3). The ANOVA and Dunnett’s multiple comparison test performed on yearly data showed a significant reduction of the number of emerged beetles in all treatments compared to the control at site 1 in 2018; treatments tefluthrin, *H. bacteriophora*, and *H. bacteriophora* + alcohol ethoxylate in 2017 at site 1; and *H. bacteriophora* + alcohol ethoxylate at site 2 in 2017 (Figure 2).

The (numerically) highest treatment efficacy, i.e., average reduction of emerged beetles in different treatments compared to control emergence, was calculated for the treatment *H. bacteriophora* + alcohol ethoxylate (49.2%), followed by treatment *H. bacteriophora* (46.2%), and tefluthrin (44.1%), which all significantly decreased the average number of emerged beetles in the three consecutive years. In the treatment with thiacloprid, an insignificant efficacy of 25.2% was observed when considering the entire three-year dataset (Table 3).

### 3.3. Evaluation of Maize Root Damage, Plant Lodging, Fresh Plant Weight, and Fresh Grain Yield Per Plant

The WCR natural population density in the soil was not high enough to cause root damage above the economic threshold. Results of the assessment of larvae damage on maize roots by the node injury scale [42] showed no statistical difference between treatments. Only two lodged plants were observed in the thiacloprid treatment at the site 1 (Bučečovci) experiment on 11 August 2016; elsewhere, no lodging was observed. In 2017, several plants exhibited typical WCR damage symptoms (i.e., ‘goose necks’), but the damage was transient as the plants recuperated. In 2018, corn lodging was not evident in the experimental plots.

The results of the general linear model and the HSD procedure showed a significant effect of treatment (F_4, 94_ = 4.27; *p* = 0.0034) and location (F_1, 94_ = 237; *p* = 0.0000) but not year (F_2, 94_ = 1.82; *p* = 0.169) or the interaction of treatment × location (F_4, 94_ = 0.11, *p* = 0.9784) on fresh plants’ weight. The lowest plant weight was measured in the negative control, followed by treatments with tefluthrin, *H. bacteriophora*, *H. bacteriophora* + alcohol ethoxylate, and thiacloprid (all treatments except tefluthrin significantly increased plant weight). Fresh grain yield per plant was affected significantly by factor location (F_1, 94_ = 161; *p* = 0.0000) and by year (F_2, 94_ = 14.99; *p* = 0.0000) but not treatment (F_4, 94_ = 2.27; *p* = 0.0689) or by the interaction of treatment × location (F_4, 94_ = 0.25, *p* = 0.907) (Table 4).

### 3.4. Assessment of Persistence of Nematodes (EPN) in the Soil

The presence of naturally occurring EPN was not confirmed. The persistence of applied EPNs was defined as the percentage of infected *G. mellonella* and *T. molitor* larvae in the tested soil samples. Results varied significantly over the period of monitoring. Throughout the duration of the experiment (2016-2018), the presence of EPNs was confirmed at both sites in all EPN-treated plots. More infected soil samples were found when taken from site 1, where, one month post-application, up to 89% of larvae used as EPN baits were infected with nematodes. With the exception of the year 2017, decreasing larval mortalities were found in later periods. However, some soil samples taken up to 141 days after the application of *H. bacteriophora* still contained virulent EPN. The highest infectivity rates were recorded in 2017 at site 1, where, 83 days after application, the addition of alcohol ethoxylate resulted in 90% mortality at 5–10 cm and 83% in soil samples taken from the 10–15-cm soil depth. On the other hand, a lower level of mortality was observed for soil samples collected at site 2, where, in 2016, the presence of EPN was confirmed only at the first sampling date, when 33% of the larvae were infected at a depth of 10–15 cm. Mixing *H. bacteriophora* solution with alcohol ethoxylate generally increased insect baits’ mortality rate in soil samples taken in 2017 and 2018 (Table 5).

## 4. Discussion

The results of the three-year field study, performed on naturally WCR-infested maize fields at two pedo-climatically different locations with dissimilar soil pest density, shows that the use of the entomopathogenic nematode *H. bacteriophora* can significantly reduce WCR adult emergence (46.2 ± 7.4% reduction), which is similar to the chemical control represented by the use of the granular soil insecticide tefluthrin (44.1 ± 11.7% reduction) (Figure 2 and Table 3). Similar findings were reported when biological and chemical-based managements were compared [28,46]. However, in contrast to our experiment, these studies were performed using artificially infested maize plants with WCR eggs, resulting in severe damage to untreated maize plants but not to the nematode-treated maize plants in which the root damage was kept below the economic thresholds [28,46]. Our study, where the natural WCR larval population in the soil did not cause significant root damage [42], did not allow us to evaluate potential lodging prevention, although it showed that *H. bacteriophora* controlled WCR larvae as well as tefluthrin, one of the commonly used chemicals for WCR management in Europe and the USA [47,48]. The data recorded in the present study for the agent tefluthrin are contrary to the observations of [49,50], where the application of tefluthrin soil insecticide made no significant impact on the emergence of WCR. Therefore, this inconsistency can be the result of precipitation on the effectiveness of soil insecticide and WCR emergence interactions, as assumed by previous research [51], and of other environmental factors, which can influence the efficacy of soil insecticide [16,52].

The efficacy of biological as well as chemical treatments has varied significantly over the years (Figure 2), as it is influenced by several environmental factors that may affect nematode or insecticide efficacy in the soil. For example, soil moisture conditions [53] and soil type [54] affect EPN infectivity and persistence in the soil. Furthermore, the characteristics of different soil types [34,36,54], as well as of soil temperatures [55,56] and the absence of larvae or alternative hosts [55,57], can impact on the efficacy of EPNs. However, such variable results are often observed in field WCR biocontrol studies [28,34,58].

The number of beetles emerging in the field cages was significantly affected by treatment as well as by year and location. The effect of ‘year’ can be attributed to the change of the field cages used at site 1 in 2018. According to [41], they affect the maize plants much more (i.e., the plants are cut at ca. 20 cm at trap set-up) than the traps used in the years 2016 and 2017, where the plants were left intact. This could mean that only the peak beetle emergence was assessed at site 1 in 2018 whereas in the trap design used in 2016 and 2017, the late emerging beetles were also caught. In addition, factor location significantly affected the number of emerging beetles. This is due mainly to different below-ground pest density and is probably connected to the field history and different climatic and edaphic conditions at the two sites. Besides the difference in soil types, the WCR was already observed at site 1 in 2004, whereas at site 2, it had been present since 2008 [45]. However, the interaction of treatment and location was insignificant, proving that the treatments gave similar results at the two locations in the three-year experiment, showing that the nematodes (and tefluthrin) performed equally well under dissimilar pest pressures and pseudo-climatic conditions, and that the two trap designs gave comparable results.

Treatments evaluated in the present study (*H. bacteriophora*, *H. bacteriophora* + alcohol ethoxylate, and thiacloprid), except that of tefluthrin, led to a significantly increased fresh plant weight. In a similar study conducted in Hungary, [46] investigated the application of entomopathogenic nematodes and a soil insecticide based on active ingredients: Tefluthrin, cypermethrin, and chlorpyrifos. In most circumstances, all these agents were found to be similarly effective on larval control, and increased the yield slightly. The application of *H. bacteriophora* in the soil successfully reduced the root damage caused by WCR larvae and by plant lodging by up to 60% [26]. Further, the application of soil insecticides to the furrows at planting can prevent larval damage, leading to yield benefits [59]. A significant positive impact of tefluthrin on the grain yield of transgenic Bt corn hybrids was reported from studies in North Dakota at a site with high WCR infestation levels [49]. In some other studies, tefluthrin did not provide a yield benefit consistently [60] but improved root protection from larval feeding injury at high larval feeding pressure [61] and significantly increased delays in WCR emergence as compared with Bt maize alone [17]. However, due to a variety of findings, treatments can be affected by biotic and abiotic factors, as discussed earlier [16,49].

Applied *H. bacteriophora* nematodes were detected almost four months after application in sandy loam soil at site 2 and almost five months after application in silty loam soil at site 1, which supports previous research [32,34,56]. These studies showed that the applied nematodes were able to survive in the soil for approximately one month without their host, which hatches 2-4 weeks after maize sowing and EPN application. They also showed that *H. bacteriophora* survived for up to 2-5 months in the soil of maize fields [56] while its persistence after application on golf courses [62] and in alfa alfa fields [63] was up to 2 years. Moreover, our experiment showed that nematodes, regardless of the soil types, reduce the emergence of WCR beetles. These results support the findings that *H. bacteriophora* has the potential to reduce WCR larvae in different soil types [34]. It was shown that the efficacy of entomopathogenic nematodes can be higher in soils with high clay and silt contents than in sandy soils [34]. This is in sync with our observations in 2017, where *H. bacteriophora* performed better at site 1 (silty loam texture) than at site 2 (sandy loam texture).

Among other abiotic factors, soil moisture content is one of the most important factors limiting the persistence and infectivity of EPNs [36], which tend to be most infective under moderate soil moisture conditions [53]. The ability of EPN to persist under longer desiccating conditions varies between species, depending on their ability to adapt metabolic activity and recover after exposure to unfavorable moisture conditions [53]. A recent study showed that combining *H. bacteriophora* with adjuvants, such as soil conditioners, may enhance the performance of nematodes against soil-dwelling insects [40]. In our study, the addition of surfactant did not increase infection rates in general, although some results obtained from site 2 in 2017 indicate that the addition of surfactant may improve the persistence and effectiveness of infective juveniles in soil compared with regular water application treatment.

One of our aims was to evaluate the efficacy of commonly used maize seeds coated with thiacloprid on WCR. Although the results showed that thiacloprid affected the number of emerged adults in the emergence cages (Figure 1 and Table 3), it did not affect WCR larvae significantly. Furthermore, it was much less effective than tefluthrin, which is commonly used against WCR larvae in Europe; therefore, we consider it an inappropriate option for WCR larvae control.

The average number of WCR beetles caught on an unbaited Pherocon AM yellow sticky trap (Trécé) in emergence cages from pest-infested soil in the control plots was approximately five-fold lower at site 2 (Šmartno pri Cerkljah, Northern Slovenia) than at site 1 (Bučečovci, Eastern Slovenia) despite the fact that both male and female beetles can be caught on unbaited yellow sticky traps. The official WCR monitoring program, however, in which PAL pheromone traps are used to capture only male beetles, has shown similar pest densities at both sites. Our study thus shows that soil pest density did not correlate with official (aerial) pest monitoring. The main reason for this lies in the pheromone’s pest attraction in the mid to long range [64]. We also assume that a certain proportion of adults could have died on the soil surface and in the maize canopy before they were caught on the unbaited yellow sticky traps in the cages, while some were washed away by the rain, decomposed, or were not stuck to the traps due to dirt accumulating on the traps. It would be better if the field cage observation interval during the peak beetle emergence was shorter. The cage catches might have been influenced by the development stage of the plant as well; if there was flowering or cobs, the adults were more attracted to plants. Nevertheless, the traps and methodology employed allowed us to successfully discriminate between the effectiveness of the different treatments investigated.

## 5. Conclusions

We conclude that WCR larvae management in maize using EPN *H. bacteriophora* could offer a sustainable WCR biological control management option in Europe. The three-year study showed that the effectiveness of chemical products against biological products varied over the years as it was affected by several abiotic factors. EPNs managed the WCR larvae population as effectively as a commonly used chemical agent tefluthrin while thiacloprid showed inconclusive results and was therefore assessed as being inappropriate for WCR larvae control. EPNs were proven to be able to persist in maize fields for almost five months regardless of dissimilar pedo-climatic conditions. The results suggested sandy loam soil to be more favorable compared to silty loam soil for EPNs treatment with alcohol ethoxylate.

## Figures and Tables

**Figure 1 insects-11-00202-f001:**
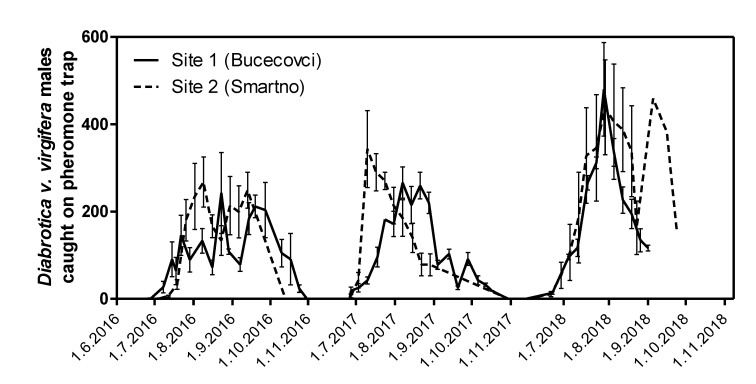
*Diabrotica v. virgifera* males caught on PAL pheromone traps 1 km away from sites 1 and 2 in 2016–2018 (*n* = 3 for site 1 and *n* = 9 for site 2).

**Figure 2 insects-11-00202-f002:**
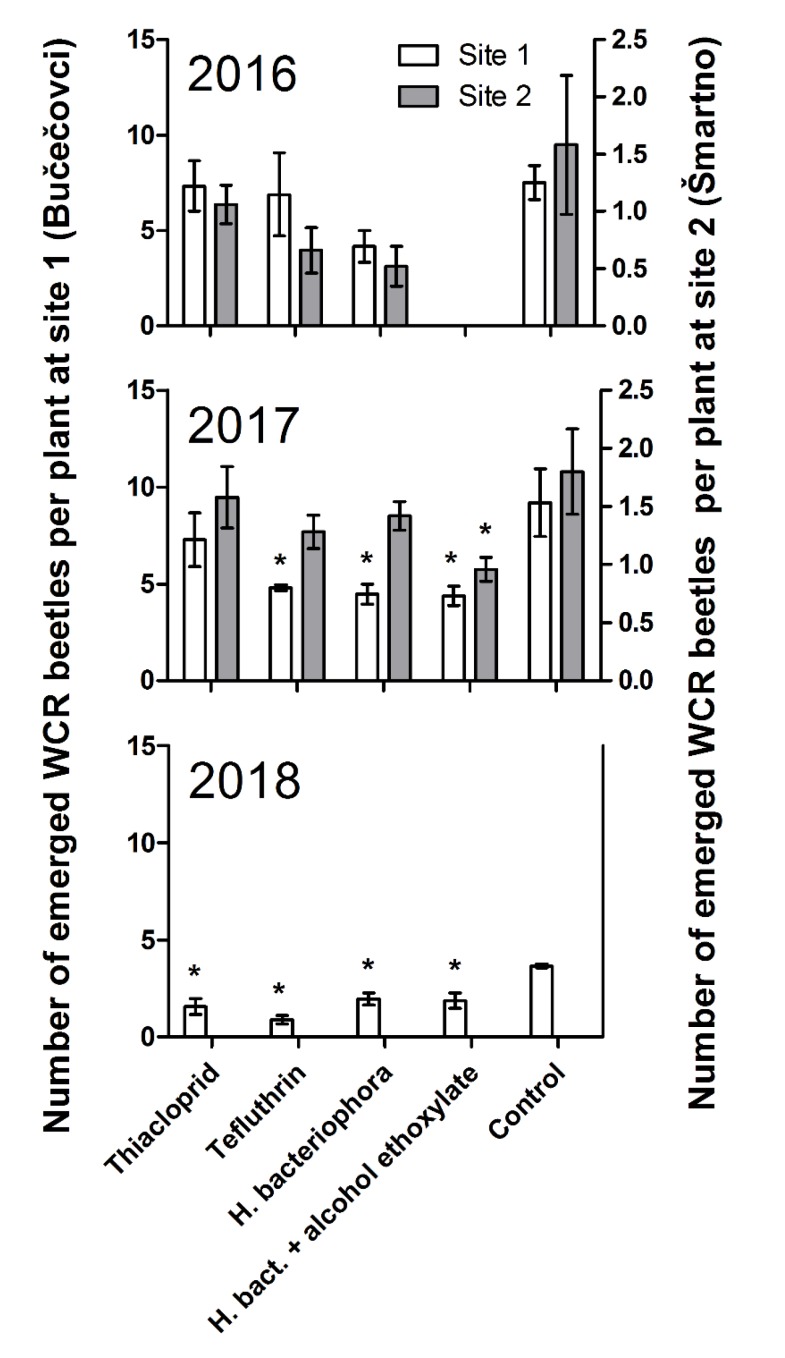
The number of western corn rootworm (WCR) beetles per plant that emerged and were caught on the yellow sticky trap in the 1-m^2^ emergence gauze cages, covering 12 plants, on both field sites in the three consecutive years (2016–2018) of field experimentation. Graphs show average and standard errors (*n* = 25 for all treatments, except 15 for *H. bact.* + alcohol ethoxylate). Asterisks indicate significant differences from the respective controls (*p* < 0.05). *H. bact.*—*H. bacteriophora*.

**Table 1 insects-11-00202-t001:** Characteristics of the field experiment.

Field	Site 1: Eastern Slovenia, Bučečovci	Site 2: Northern Slovenia, Šmartno Pri Cerkljah
Experiment location	46°35’07"N 16°06’37"E	46°15’08.8"N 14°29’54.7"E
Date of sowing/nematode application	22 April 2016	11 May 2016
	26 April 2017	17 May 2017
	9 May 2018	
Maize seed	Chapalu (Agrosaat)	LG 34.90 (Agrosaat)
Field size	0.14 ha	0.14 ha
Soil types	silty loam texture	sandy loam texture

**Table 2 insects-11-00202-t002:** Western corn rootworm management on two field sites in three consecutive years (2016–2018). Site 1—Bučečovci E Slovenia; site *2*—Šmartno pri Cerkljah, N Slovenia.

Field Site	Date of Sowing and Application	Treatment Active Ingredient (Tradename)	Dose
1	22 April 2016	1. Untreated maize seed (Control)	85,000 plants/ha
		2. *H. bacteriophora* (Dianem)	2 × 10^9^ nematodes/ha with 200 L water
		3. Tefluthrin (Force 1,5 G)	12 kg/ha (1.5% a.i.)
		4. Thiacloprid (Sonido FS 400)	0.125 L/50,000 seeds (40% a.i.)
	26 April 2017	1. Untreated maize seed (Control)	85,000 plants/ha
		2. *H. bacteriophora* (Dianem)	2 × 10^9^ nematodes/ha with 200 L water
		3. Tefluthrin (Force 1,5 G)	12 kg/ha (1.5% a.i.)
		4. Thiacloprid (Sonido FS 400)	0.125 L/50,000 seeds (40% a.i.)
		5. *H. bacteriophora* + alcohol ethoxylate (Dianem + Transformer)	5 L/ha, 20% w/w
	9 May 2018	1. Untreated maize seed (Control)	85,000 plants/ha
		2. *H. bacteriophora* (Dianem)	2 × 10^9^ nematodes/ha with 200 L water
		3. Tefluthrin (Force 1,5 G)	12 kg/ha (1.5% a.i.)
		4. Thiacloprid (Sonido FS 400)	0.125 L/50,000 seeds (40% a.i.)
		5. *H. bacteriophora* + alcohol ethoxylate (Dianem + Transformer)	5 L/ha, 20% w/w
2	11 May 2016	1. Untreated maize seed (Control)	85,000 plants/ha
		2. *H. bacteriophora* (Dianem)	2 × 10^9^ nematodes /ha with 200 L water
		3. Tefluthrin (Force 1,5 G)	12 kg/ha (1.5% a.i.)
		4. Thiacloprid (Sonido FS 400)	0.125 L/50,000 seeds (40% a.i.)
	17 May 2017	1. Untreated maize seed (Control)	85,000 plants/ha
		2. *H. bacteriophora* (Dianem)	2 × 10^9^ nematodes /ha with 200 L water
		3. Tefluthrin (Force 1,5 G)	12 kg/ha (1.5% a.i.)
		4. Thiacloprid (Sonido FS 400)	0.125 L/50,000 seeds (40% a.i.)
		5. *H. bacteriophora* + alcohol ethoxylate (Dianem + Transformer)	5 L/ha, 20% w/w

**Table 3 insects-11-00202-t003:** Overall treatment efficacy calculated from the three-year dataset accumulated on two experimental sites. Efficacy is expressed as the average reduction of beetles that emerged in different treatments compared to the control emergence (%). *n* = 25 for all treatments, except 15 for *H. bacteriophora* + alcohol ethoxylate. Columns not sharing the same letters are significantly different. *SE*—standard error; *nd*—no data.

Year	Site/Treatment	Thiacloprid	Tefluthrin	*H. bacteriophora*	*H. bacteriophora* + Alcohol Ethoxylate	Control
2016	1	2.7%	8.4%	44.7%	nd	0.0%
	2	33.0%	58.5%	67.0%	nd	0.0%
2017	1	20.9%	47.7%	51.5%	52.5%	0.0%
	2	12.0%	29.6%	21.3%	46.3%	0.0%
2018	1	57.2%	76.0%	46.6%	48.8%	0.0%
Average efficacy ± SE		25.2 ± 9.4%	44.1 ± 11.7%	46.2 ± 7.4%	49.2 ± 1.8%	0.0 ± 0.0%
HSD Multiple Range test		AB	A	A	A	B

**Table 4 insects-11-00202-t004:** Effect of different treatments on fresh plant weight and fresh grain yield per plant. Yield was assessed by weighing ears (fresh, with husks) from five plants. Data presented are averages ± standard error (*n* = 20 for all treatments, except *n* = 15 for *H. bacteriophora* + alcohol ethoxylate). Treatments not sharing the same letters are significantly different.

Treatment	Weight of 1 Maize Plant [g]			Fresh Grain Yield Per Plant [g]	
	Site 1	Site 2	HSD test	Site 1	Site 2
**Thiacloprid**	557 ± 36	1033 ± 36	A	246 ± 13	395 ± 13
**Tefluthrin**	490 ± 36	956 ± 36	AB	222 ± 13	365 ± 13
***H. bacteriophora***	523 ± 36	1021 ± 36	A	240 ± 13	382 ± 13
***H. bacteriophora* + alcohol ethoxylate**	561 ± 36	1011 ± 36	A	259 ± 13	385 ± 19
**Control**	440 ± 36	1021 ± 36	B	216 ± 13	369 ± 13

**Table 5 insects-11-00202-t005:** Days of entomopathogenic nematode persistence at different depths (cm), expressed as the percentage of infected insect cadavers used as insect baits. Larvae of *T. molitor* and *G. mellonella* were used as live bait for entomopathogenic nematodes and placed on soil samples collected at two depths at both experimental sites; nd–no data.

Site/Year	Days Post Treatment	Treatment/Depth of Sampling [cm]			
		*H. bacteriophora*		*H. bacteriophora* + Alcohol Ethoxylate	
		5–10	10–15	5–10	10–15
**Site 1**					
2016	26	78	89	nd	nd
	61	11	22	nd	nd
	83	11	33	nd	nd
2017	30	73	37	50	43
	57	0	0	10	37
	83	50	63	90	83
	113	8	8	25	13
	141	4	8	4	0
2018	20	13	27	40	27
	41	13	7	7	13
	57	0	0	0	22
	69	0	7	0	0
	83	13	13	3	3
	100	13	0	0	17
	128	0	0	13	0
**Site 2**					
2016	28	0	33	nd	nd
	49	0	0	nd	nd
	84	0	0	nd	nd
2017	22	4	0	7	11
	57	33	10	60	50
	114	8	4	13	8

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
