# Peer review of "Evaluation of the Field Efficacy of Heterorhabditis Bacteriophora Poinar (Rhabditida: Heterorhabditidae) and Synthetic Insecticides for the Control of Western Corn Rootworm Larvae"

_insects, 2020, doi:10.3390/insects11030202_

Round 1

Reviewer 1 Report

Dear Authors, 

The investigation/improvements of knowledge using biological control agents against pest insects on field scale is important. The literature research was broad and introducing the topic well. Methods used are common and the results are also discussed in detail. 

My comments are in detail: 

Author Response

Dear reviewer 1

Reviewer 2 Report

The paper entitled «evaluation of field efficacy of h bact and synthetic insecticides for the control of wcr larvae» is an interesting comparison of biological and chemical treatments. As wcr larvae are difficult to manage, and options become less and less, this study is worth to be published after a suggested minor revision. General comments: The manuscript seems to have been written in a rush. There are a number of typos, words missing, sentences mixed up, etc. It is not clear whether the exactly same fields were used over years. And if yes, can the treatments not have an effect on the following year? e.g. EPNs? For better comparison among trials, please convert adult emergence data to adult emergence per plant and biomass to biomass per plant. In text and tables and graphs. I suggest calculating the accumulated pheromone trap captures per year and then to calculate % of accumulated captures per time step. Then you could pool all 3 years, and provide a more general info for that region of Slovenia. Or you take proportions of max. Details: L 4 … and synthetic… Abstract Pls avoid abbreviations, WCR, EPN, etc. L16 I am not sure it is important in eastern Europe. L18 to 24 Please clarify how many fields and year, thus replications per treatment L26 Suggest deleting sentence «They exhibit….» L27 Pls clarify low and high pressure. Pls clarify the message, e.g. do treatments work well at low or medium or high pest densities, or what? L32 emerging…. what? L34 sentence not clear. soil? L27 to 36 please provide some average efficacy numbers for control of wcr and for preventing root damage L113 how distance are fields to each other? L115 not clear when were the monoculture periods if the beetles were in the fields since 2004 and 2008. Why are there beetles if there was no monoculture all the time? L116 -117 move this info to L125 Table 1 Are the 2 columns every year exactly the same field? If yes, can the treatments not affect the following year? Why so late planting in field cerkljah? L120 … inter-plant spacing of… L121 … resulting in 85000 plants per ha L123 please clarify replicate numbers per treatments Do you mean «5 plots of 4 rows x 100 m for each treatment per each field»? L127 were did you place those traps? into the untreated control? Table 2 Avoid abbreviations in captions L131 Delete sentences on a.i. and put this into column heads of table Column heads, e.g. Field site Date of sowing and application Treatment active ingredient (tradename) Dose Put a.i. name of transformer into correct column L140 … batches through assessing… L152 a.i. of transformer needs to be mentioned here L161 to L164 is this only relevant for the untreated control? I have the feeling maize hybrid, bird repellents, fungicides are on all treatments, or not? If not, then this needs to be explained why not. L173 How many plants do the Rauch cages cover? L175 For better comparison among trials, please convert adult emergence data to adult emergence per plant. In text and tables and graphs. L209 i am not sure about LSD test. As far as i know it does not adjust for multiple comparison. Why do you not use Dunnet for all the comparisons? L216 to 221 For better comparison among trials, please convert adult emergence data to adult emergence per plant. In text and tables and graphs. L222 why «in contrast». It is not clear where those traps were placed? Pheromone traps can attract from medium long distance. L225 each year of the experiments or each year in general in that region. If the latter, then this is discussion. Figure 1 caption needs to mention that those dvv are only males. And where were the traps placed? Into untreated control? I suggest calculating the accumulated captures per year and then to calculate % of accumulated captures per time step. Then you could pool all 3 years, and it would give you a more general info for that region of Slovenia. Or you take proportions of max. L132 Start with: General…… showed….. L234 Mention please what was the general level of control, 44 to 49%??? L235 and 236 Write that there was no difference detected between untreated control and thiacloprid L238 i think, you need to be careful with LSD test Figure 2 Please put all y axis onto same scale. Currently it is very confusing. Please standardise data of dvv emerged per plant. Per cage does not help readers L244 no abbreviations lease in caption L245 …emergence gauze cages…. how many cages per how many plots per treatment ? L250 to 254 there is no highest or lowest if there was no difference detected. I suggest deleting this text block Table 3 Maybe you could delete last column I do not think LSD is appropriate as it does not account for multiple comparison. It is just many times doing t-test Suggest deleting row of 2018 field 2, or why is it shown? Table 4 please convert to gram/plant yields are what, kg per ha? please reduce decimals Table 5 avoid abbreviations how many of the cases with t molitor, how many with galleria? Header in table « Days persistence at different depths (cm)» Please add rows on averaging fields

Author Response

Dear reviewer 2

Reviewer 3 Report

The manuscript by Modic et al. evaluates the field efficacy of Heterorhabditis bacteriophora nematodes against western corn rootworm. The work is well described and scientifically sound. Overall the paper is well-written and the authors are careful in their analyses, there are a few areas where the manuscript can be remedied by textual modifications. It was a pleasure to read this particular paper and I congratulate the authors on a nice contribution to the field. I especially appreciated some of the careful wording such as the sentence “The presence of naturally occurring EPN was not confirmed.” Well written.

Minor suggestions:

32: change to “beetles emerging”

34: this line is incomplete and ends with “, soil type.” The authors should complete the thought here to finish the sentence.

52-53: This sentence needs to be rewritten. “and through cross-resistance,” doesn’t make sense.

100: I suggest they use the word “soil” instead of “edaphic.” While “edaphic” is correctly used here, it is not a very well-known word.

144: Change to “The IJs were applied…”

145: Change to “IJs per ha in 200…”

146-147: For the special EPN application system, a picture of the device, even as a supplemental figure, would be helpful to clarify the method.

229-230: Add in a line that this data is for control fields/plots.

236: The oft-repeated statement that the thiacloprid treatment was insignificant is a little confusing and should be clarified. Perhaps from table 3 which averages the data over the 3 years the treatment is insignificant, but the data shown in figure 2 makes clear that the thiacloprid treatment is significantly different from the control in 2018. So the authors should clarify this statemen, which is repeated throughout the manuscript.

237-238: Again, the authors state that the decrease in beetle emergence in the tefluthrin, Hb, and Hb + soil conditioner were indistinguishable. They need to clarify by stating that the “average decrease in beetle emergence over the 3 years…” is indistinguishable, because the 2018 data suggest that they are distinguishable.

254: Again, they need to clarify that the average effect of thiacloprid was not significant, since the 2018 data are significant.

284: remove the second varied and the phrase “of EPN persistence monitoring.”

285: change to “of EPNs was confirmed at both sites in all treated plots.”

297-298: fix the spacing issue with the reference to table 5.

305: Change to “use of the entomopathogenic…”

306: Change to “which is similar to the chemical…”

317: Adjust the phrasing since in 2018 the use of tefluthrin did make an impact on WCR emergence.

324: Remove the word “its.”

328: Change to “The number of beetles… was significantly affected…”

343: Change to “led to significantly increased fresh plant weight.”

359: Change to “which supports previous research…”

Author Response

Dear reviewer 3
